# A Review of Radiomics and Artificial Intelligence and Their Application in Veterinary Diagnostic Imaging

**DOI:** 10.3390/vetsci9110620

**Published:** 2022-11-08

**Authors:** Othmane Bouhali, Halima Bensmail, Ali Sheharyar, Florent David, Jessica P. Johnson

**Affiliations:** 1High Energy and Medical Physics Group, Department of Engineering, Education City, Texas A&M University at Qatar, Doha P.O. Box 23874, Qatar; 2Qatar Computing Research Institute, Department of Sciences, Education City, Hamad Bin Khalifa University, Doha P.O. Box 34110, Qatar; 3Equine Veterinary Medical Center, A Member of Qatar Foundation, Al Shaqab Street, Al Rayyan, Doha P.O. Box 6788, Qatar

**Keywords:** radiomics, artificial intelligence, veterinary medicine, diagnostic imaging

## Abstract

**Simple Summary:**

The goal of this paper is to provide an overview of current radiomic and AI applications in veterinary diagnostic imaging. We discuss the essential elements of AI for veterinary practitioners with the aim of helping them make informed decisions in applying AI technologies to their practices and that veterinarians will play an integral role in ensuring the appropriate uses and suitable curation of data. The expertise of veterinary professionals will be vital to ensuring suitable data and, subsequently, AI that meets the needs of the profession.

**Abstract:**

Great advances have been made in human health care in the application of radiomics and artificial intelligence (AI) in a variety of areas, ranging from hospital management and virtual assistants to remote patient monitoring and medical diagnostics and imaging. To improve accuracy and reproducibility, there has been a recent move to integrate radiomics and AI as tools to assist clinical decision making and to incorporate it into routine clinical workflows and diagnosis. Although lagging behind human medicine, the use of radiomics and AI in veterinary diagnostic imaging is becoming more frequent with an increasing number of reported applications. The goal of this paper is to provide an overview of current radiomic and AI applications in veterinary diagnostic imaging.

## 1. Introduction

Human health care has shown great advances in the application of radiomics and artificial intelligence (AI) in a variety of areas, ranging from hospital management and virtual assistants to remote patient monitoring and medical diagnostics and imaging [1]. Disciplines dealing with large data components, in particular, benefit from the assistance of AI. Digital imaging is used in a wide variety of clinical settings, in both human and veterinary medicine, including X-ray, ultrasound (US), computed tomography (CT), magnetic resonance imaging (MRI), positron emission tomography (PET) scans, retinal photography, dermoscopy, and histology, among others [1,2]. These fields already benefit from the applications of radiomics and AI [1,2]. The combination of radiomics and AI with diagnostic imaging offers a number of advantages, including their automated ability to perform complex pattern recognition, which can be applied in an accurate and reproducible manner to digital image analysis [1]. This can be integrated into a variety of medical imaging applications, including disease detection, characterization, and monitoring. Traditionally, this role is performed by trained diagnostic imaging specialists who assess and evaluate medical images. However, image interpretation can be subjective and greatly influenced by education and prior experience. Consequently, there has been a recent move to integrate them as tools to assist clinical decision making and to incorporate it into routine clinical workflows and diagnoses. However, lagging behind human medicine, the use of radiomics and AI in veterinary diagnostic imaging is becoming more frequent with an increasing number of reported applications. The goal of this paper is to provide an overview of current AI applications in veterinary diagnostic imaging.

## 2. Radiomics and AI

### 2.1. Radiomics

In medicine, various ways to generate big data exist, including the well-known fields of genomics, proteomics, metabolomics, transcriptomics, and microbiomics. Imaging is increasingly being utilized to build a specific omics cluster called “radiomics”, which is similar to these “omics” clusters. Radiomics is a relatively recent area of precision medicine [3]. It is a quantitative approach to medical imaging that tries to improve the data available to doctors through advanced and sometimes counterintuitive mathematical analysis [3]. The notion of radiomics is founded on the assumption that biological images contain information about disease-specific processes that is undetectable to the human eye and thus unavailable through typical visual analysis of the generated image [4]. It consists of the extraction of a large number of features from medical images and modalities. The mapping of these images into quantitative data, which can be mined with appropriate and sophisticated statistical tools, can be an important step toward personalized precision medicine (PPM) [5,6,7]. Radiomics has been applied to different fields in human health, including magnetic resonance imaging (MRI), computed tomography (CT), positron emission tomography (PET), and ultrasound (US) [8,9,10,11].

Traditionally, information from medical images was extracted solely based on visual inspection. In addition to the intra- and inter-operator variability, visual extraction cannot pull out important information hidden in these images. Radiomics was introduced as a tool to extract as much quantitative hidden information as possible to aid practitioners in diagnosis and decision making. As an example, in their study of tumor phenotype for lung and head-neck cancer, and using CT images with a radiomic approach, Aerts et al. (2014) found several new features that were not identified as significant previously, with high prognostic power [12]. This radiomic signature was validated with biological data on three independent data sets, demonstrating the performance of radiomics [12].

The workflow of radiomics is usually described in four steps: (i) preprocessing, (ii) imaging and image segmentation, (iii) feature extraction such as shape, texture, intensity, and filters, and (iv) feature analysis (see Figure 1) [13]. In the following, we briefly describe each of these four steps.

*Preprocessing*: Image preprocessing is the practice of applying a series of transformations to an initial image in order to improve image quality and make statistical analysis more repeatable and comparable. However, there is no predetermined analytic method to do this, and it varies depending on collected data and disease to study. Since radiomics deals primarily with images, it depends highly on image parameters of a given modality such as: the size of pixels (2D) or voxels (3D), the number of gray levels and range of the gray-level values, as well as the type of 3D-reconstruction algorithms [14]. This is important because the stability and robustness of radiomics features depend on the image processing settings [14,15]. Image acquisition, segmentation, intensity normalization, co-regulation, and noise filtering are three more analytical techniques that are crucial for processing the quantitative analysis of the images (Figure 2).*Segmentation*: This is the delineation of the region of interest (ROI) in 2D or volume of interest (VOI) in 3D. This is the most critical step in the radiomics workflow since it specifies the area/volume from which the features will be extracted. Segmentation is tedious and is usually done manually by a human operator or semi-manually using standard segmentation software [16]. However, segmentation is subject to intra- and inter-operator variability. Therefore, fully automated segmentation was introduced recently, using deep learning techniques [17]. Deep learning (DL) is based on artificial neural networks where multiple layers (such as neurons in the human brain) are used to perform complex operations and extract higher levels of features from data. DL techniques require a large amount of data and considerable computing resources to achieve the required accuracy.*Feature extraction*: This is mainly a software-based process aimed at extracting and calculating quantitative feature descriptors. Most of the feature extraction procedures follow the guidelines of ISBI “Image Biomarker Standard Initiative”, which clusters features in major categories such as intensity-based features, shape and edge features, texture features, and morphological features [18]. On the other hand, using only gray-level descriptors or histograms provides no information on the spatial distribution of an image’s content, which can be obtained by evaluating texture features [19]. Because of their varied textures, regions with comparable pixels/voxels can be distinguished in some images. Because texture features can represent the intricacies of a lesion observed in an image, they have become increasingly relevant [20]. The geometric features extracted from the segmented object, such as its contours, junctions, curves, and polygonal regions, are referred to as shape features. Quantifying item shapes is a difficult task because it is dependent on the efficacy of segmentation algorithms. Moreover, methods such as wavelet transformation, Laplacian of Gaussian, square root, and local binary pattern are used to polish this step. For example, a wavelet transformation is a powerful tool for multiscale feature extraction. A wavelet is a function that resembles a wave. Scale and location are the basic characteristics of the wavelet function. The scale defines how "stretched" or “squished” the function is, whereas placement identifies the wavelet’s position in time (or space). Decreasing the scale (resolution) of the wavelet can capture high-frequency information and, therefore, analyze well high-spatial frequency phenomena localized in space and can effectively extract information derived from localized high-frequency signals. Laplacian of Gaussian filter smooths the image by using a Gaussian filter, then applies the Laplacian to find edges (areas of gray-level rapid change). Square and square root image filters are tagged as Gamma modifiers. The square filter is accomplished by taking the square of image intensities, and the square root filter by taking the square root of the absolute value of image intensities [21]. Local binary pattern relies on labeling a binary value to each pixel of the image by thresholding the neighboring pixels based on the central pixel value, and the histogram of these labels is considered as texture features [22].*Feature analysis*: The number of features extracted can be very high, which makes the analysis process cumbersome and the application of artificial intelligence ill-posed, in particular, if the number of data is not high. Reducing the number of features to a reasonable yet meaningful number is called “feature selection” or “dimension reduction” and helps to exclude features that are redundant and non-relevant from the data set before doing the final analysis. It also helps gather only the features that are the most consistent and relevant to build a reliable model for further prediction and classification [23]. Dimension reduction techniques, such as principal component analysis and partial least squares, construct ‘super variables’—usually linear combinations of original input variables—and use them in classification. Although they may also lead to satisfactory classification, biomedical implications of the classifiers are usually not obvious since all input features are used in the construction of the super features and hence classification. Feature selection methods can be classified into three categories. The filter approach separates feature selection from classifier construction. This implies that the machine learning algorithm handles the feature removal and data classification in separate steps. As a result, the algorithm begins by picking out the most crucial features and eliminating the others, and then, in the second step, it only uses those features to classify the data. The wrapper approach measures the “usefulness” of features based on the classifier performance by using a greedy search approach that evaluates all the possible combinations of features against the classification-based evaluation criterion and keeps searching until a certain accuracy criterion is satisfied. The embedded approach embeds feature selection within classifier construction. Embedded approaches have less computational complexity than wrapper methods. Compared with filter methods, embedded methods can better account for correlations among input variables. Penalization methods are a subset of embedded methods in which feature selection and classifier construction are achieved simultaneously by computing parameters involved in the penalized objective function. Many algorithms have been proposed to achieve this; the most popular ones are lasso, adaptive lasso, bridge, elastic net, and SCAD, to name a few [24,25,26,27,28]. Table 1 summarizes an assessment of some publicly available open-source radiomics extraction tools and their primary characteristics.

### 2.2. AI

Artificial Intelligence (AI) has attracted a great deal of attention in the past decade. It is a field that uses and develops computer systems to imitate human intelligence and learn from experience to perform and improve the tasks assigned. Machine Learning (ML) is a major subfield of AI that develops algorithms to learn from existing data and perform statistical inference to make accurate predictions of new data. The training of the algorithm on the data can be performed in two ways: supervised and unsupervised. Supervised learning trains its algorithms on previously labeled/annotated data to find the relationship between the labels and the data features and generalizes this knowledge to predict new (unlabeled) cases. Unsupervised learning (also known as self-supervised learning) refers to the process of grouping data into clusters using automated methods or algorithms on data that has not been classified or categorized and finding the relationship between intrinsic features to categorize the data into clusters (see Figure 3) [34]. Many machine learning models are linear and, therefore, cannot capture all features that are intrinsically nonlinear. Several machine learning algorithms based on nonlinear models have arisen in recent decades to solve regression, classification, and estimation challenges. A linear model for prediction uses a linear function, whereas a nonlinear model uses a nonlinear function coupled with computational complexity (which limits its use). In classification, linearity refers to the fact that the decision surface is a linear separator, such as a line that divides positive and negative points in the training set in the case of a plane. Similarly, a nonlinear model will use a nonlinear decision surface, such as a parabola, to divide classes. Details of linear and nonlinear learning models are also summarized in Table 2.

Deep learning (DL) is a more advanced subfield of ML. Instead of teaching the algorithm to process and learn from the data, a DL algorithm teaches itself to process and learn from the data. This is done through layers of artificial neural networks (ANN) using large amounts of data (see Section 2.3) [51]. Another type of machine learning algorithm is called reinforcement learning, where the algorithm is trained to take a sequence of decisions based on trial and error in which, after each operation, the algorithm gets rewarded or penalized until a solution is achieved [52]. There are several metrics to assess the outcome of AI models. These are sensitivity, specificity, and accuracy. Sensitivity is the proportion of positive cases (e.g., malignant tumors) that are reported as positive cases. Specificity is the proportion of negative cases that are reported as negative cases. Accuracy is the proportion of all correct cases that are reported as correct (either negative or positive) cases. In most applications in medicine, supervised learning is the preferred strategy for association/prediction or classification, particularly in radiomics. It applies the same concept as a student learning under the supervision of the teacher. Figure 4 summarizes the principle of employing supervised learning techniques from creating, training, and testing data to prediction or classification.

### 2.3. Radiomics-AI Combination

The field of radiomics deals with an ever-growing number of images and imaging modalities. The resulting excessive number of features extracted from these images requires sophisticated and powerful data analytic tools beyond traditional statistical inference, which only AI can provide. Moreover, these features provide valuable quantitative metrics that are perfectly suited for AI algorithms. Consequently, the fields of radiomics and AI became easily interchangeable in diagnostic imaging [53]. With the development of artificial intelligence and new machine learning tools, auxiliary diagnostic systems have expanded greatly and have been used in many different tasks with all medical imaging modalities. One of the areas of artificial intelligence that has been gaining attention in the scientific community most recently is deep learning [54]. Traditional machine learning methods have limitations in data processing, mainly related to the need for segmentation and the development of feature extractors to represent images and serve as input for the classifiers. Therefore, researchers began to develop algorithms that integrated the processes of feature extraction and image classification within the ANN itself. Therefore, in deep learning techniques, the need for preprocessing or segmentation is minimized. However, the method also has disadvantages, such as the need for a very large set of images (hundreds to thousands), greater dependence on exam quality and clinical data, and difficulty in identifying the logic used (“processing black box”). The most widely known method of deep learning in medicine is that involving a convolutional neural network (CNN). A CNN is basically composed of three types of layers: the first (convolutional layer) detects and extracts features; the second (pooling layer) selects and reduces the amount of features; and the third (fully connected layer) serves to integrate AI features extracted by the previous layers, typically by using a multilayer perceptron-like neural network to perform the final image classification, which is given by the prediction of the most likely class [55,56].

### 2.4. Validation

Another important step in the machine learning process is validation and performance assessment. Given a set of images, a machine learning classifier must use at least two different subsets to perform algorithm training and predictive model validation. A widely used strategy in radiomics is cross-validation. In cross-validation, the samples are separated into N subsets: one for training, one for validation, and one (independent subset) for testing only [57,58]. Another strategy is K-fold validation, which is based on dividing data into K subsets: one for training, one for validation, and one for test and shuffling randomly this process K times. Performance is typically evaluated by calculating the accuracy, sensitivity, specificity, and area under the receiver operating characteristic (ROC) curve for the method in question. An area under the curve (AUC) closer to 1 (on a scale from 0 to 1) indicates greater accuracy of the method

### 2.5. Open-Source Data for Radiomics

Publicly accessible data sets, such as the RIDER data set, aid in the understanding of the impact of various parameters in radiomics [59]. Furthermore, the availability of a public phantom data set for radiomics reproducibility tests on CT could aid in determining the impact of collection parameters in order to minimize non-robust radiomic characteristics [60]. However, more research is needed to see if data collected on a phantom can be used on humans [61]. Similar endeavors for PET and MRI would aid in the knowledge of how changes in environments affect radiomics. To put it another way, open-source data is critical to the future advancement of radiomics.

## 3. Application of AI and Radiomics in Veterinary Diagnostic Imaging

### 3.1. Lesion Detection

One of the earliest publications describing the use of AI in veterinary diagnostic imaging involved the evaluation of a linear partial least squares discriminant analysis (PLS-DA) and a nonlinear artificial neural network (ANN) model in the application of machine learning for canine pelvic radiograph classification (see Table 3) [62]. Classification error, sensitivity, and specificity of 6.7%, 100%, and 89% for the PLS-DA model and of 8.9%, 86%, and 100% for the ANN model were achieved [62]. Although the classification in this study was not focused on the presence of hip joint pathology but on the presence of a hip joint in an image, this study was one of the first to demonstrate that common machine learning algorithms could be applied to the classification of veterinary radiographic images and suggested that for future studies the same models could potentially be used for multiclass classifiers [62]. 

Yoon and colleagues (2018) were among the first to perform a feasibility study that evaluated bag-of-features (BOF) and convolutional neural networks (CNN) in veterinary imaging for the purpose of computer-aided detection to identify abnormal canine radiographic findings, which included cardiomegaly, abnormal lung patterns, mediastinal shift, pleural effusion, and pneumothorax (see Table 3) [66]. The results indicated that while both models showed the possibility of improving work efficiency with the potential for double reading, CNN showed higher accuracy (92.9–96.9%) and sensitivity (92.1–100%) when compared to BOF (accuracy 74.1–94.8%; sensitivity 79.6–96.9%) [66]. Later, Boissady et al. (2020) developed a unique deep neural network (DNN) for thoracic radiographic screening in dogs and cats for 15 different abnormalities [65]. For the purpose of training, more than 22,000 thoracic radiographs, with corresponding reports from a board-certified veterinary radiologist, were provided to the algorithms. Following training, 120 radiographs were then evaluated by three groups of observers: the best-performing network, veterinarians, and veterinarians aided by the network. The results showed that the overall error rate of the network alone was 10.7%, significantly lower (*p* = 0.001) than the overall error rate of the veterinarians (16.8%) or the veterinarians aided by the network (17.2%) [65]. It is interesting to note that the network failed to statistically improve the veterinarians’ error rate in this study, which the authors hypothesized could be due to a lack of experience with the use of AI as an aid and failure to trust CNN’s pattern recognition [65]. These results indicated that although the network could not provide a specific diagnosis, it could perform very well at detecting various lesion types (15 different abnormalities), confirming the usefulness of CNN for the purpose of identification of thoracic abnormalities in small animals [65].

Several papers have been published investigating the use of AI applied to rodents, which are commonly used as animal models of disease (see Table 3). In these cases, the studies focus on liver disease. In 2018, one such study investigated whether AI could be used to detect texture features on mouse MRIs, which could be correlated with metastatic intrahepatic tumor growth before they become visible to the human eye [64]. The results of this study suggested that livers affected by both neoplastic metastases and micrometastases develop systematic changes in texture features [64]. Three clusters or features derived from each of the gray-level matrices were found to have an independent linear correlation with tumor growth [64]. The authors concluded that changes in texture features at a sub-resolution level could be used to detect micrometastases within the liver before they become visually detectable by the human eye [64]. 

Another report on the use of radiomics in veterinary medicine describes a radiomics-based approach to the analysis of micro-CTs (μCT) of equine proximal sesamoid bones to be able to distinguish image features in controls compared to cases who developed catastrophic proximal sesamoid bone fractures (see Table 3) [63]. Using radiomics, it was possible to consistently identify differences in image features between cases and controls, as well as highlight several features previously undetected by the human eye [63]. This work provides an initial framework for future automation of image biomarkers in equine proximal sesamoid bones, with potential applications including the identification of racehorses in training at high risk of catastrophic proximal sesamoid bone fracture [63]. 

### 3.2. Lesion Characterization

The term ‘characterization’ of a lesion encompasses the segmentation, diagnosis, and staging of a disease [1]. This depends on a number of quantifiable radiological characteristics of a lesion, including size, extent, and texture [1]. Humans are limited in their capability to interpret medical diagnostic images in this regard due to our finite capacity to handle multiple qualitative features simultaneously. AI, on the other hand, has the capacity to process a large number of quantitative features in a reproducible manner. In the veterinary literature, several examples exist of the use of AI for lesion characterization in a variety of applications. 

Interpretation and characterization of brain lesions on MRI can be challenging. In 2018, Banzato et al. evaluated the ability and accuracy of a deep CNN to differentiate between canine meningiomas and gliomas on pre- and post-contrast T1-weighted and T2-weighted MRI images and developed an image classifier based on this to predict whether a lesion (characterized by final histopathological diagnosis) is a meningioma or glioma [67]. The image classifier was found to be 94% accurate on post-contrast T1-weighted images, 91% on pre-contrast T1-weighted images, and 90% on T2-weighted images, thus concluding that it had potential as a reliable tool to distinguish canine meningiomas and gliomas on MRIs [67]. 

Similarly, more recently, the use of Texture Analysis (TA) to differentiate canine glial cell neoplasia from noninfectious inflammatory meningoencephalitis was investigated [70]. This can be challenging even for experienced diagnostic imaging specialists due to a number of overlapping image characteristics. A group of 119 dogs with diagnoses confirmed on histology were used, 59 with gliomas and 60 with noninfectious inflammatory meningoencephalitis [70]. The authors found that cohorts differed significantly in 45 out of 120 texture metrics [70]. TA was unable to classify glioma grade or cell type correctly and could only partially differentiate between subtypes of inflammatory meningoencephalitis (e.g., granulomatous vs. necrotizing) [70]. However, with a random forest algorithm (supervised learning algorithm where the "forest" built is an ensemble of decision trees, usually trained with the “bagging” method), its accuracy for differentiating between inflammatory and neoplastic brain disease was found to approach that previously reported for subjective radiologist evaluation [70].

Another study focusing on rodent livers described the use of quantitative analysis of computer-extracted features of B-mode ultrasound as an alternative non-invasive method to liver biopsy for the characterization of hepatic fibrosis [68]. Computer-extracted quantitative parameters included brightness and variance of the hepatic B-mode ultrasounds [68]. Hepatic fibrosis induced in rats (*n* = 22) through oral administration of diethylnitrosamine (DEN) showed an increase in hepatic echo intensity from 37.1 ± 7.8 to 53.5 ± 5.7 (at 10 weeks) to 57.5 ± 6.1 (at 13 weeks), while the control group remained unchanged at an average of 34.5 ± 4 [68]. A similar effect was seen over time in the hepatorenal index, heterogeneity, and anisotropy. Three other features were studied that also increased over time in the DEN group44. Subsequent hepatic histology revealed more severe fibrosis grades in DEN rats compared to controls [68]. The results showed that increasing parameters in US showed a significant positive correlation with increasing fibrosis grades, with anisotropy having the strongest correlation (*p* = 0.58) [68]. Computer-extracted features of B-mode US images consistently increased over time in a quantifiable manner as hepatic damage and fibrosis progressed in rats, making this quantitative tool a potentially beneficial adjunct to the clinical diagnosis and assessment of hepatic fibrosis and chronic liver disease [68].

CNN technology has also been developed to classify canine corneal ulcer severity (normal vs. superficial vs. deep) based on corneal photographs, which had previously been classified by veterinary ophthalmologist evaluation [69]. Following labeling and learning of images (1040 in total), they were then evaluated using GoogLeNet [71], ResNet [72], and VGGNet [73] models to determine the severity, using simulations based on an open-source software library, which was fine-tuned using a CNN model trained on the ImageNet data set [69]. Accuracies greater than 90% were achieved for most of the models for the classification of superficial and deep corneal ulcers, with ResNet [72] and VGGNet [73] achieving accuracies >90% [69]. This study concluded that the proposed CNN method could effectively differentiate ulcer severity in dogs based on corneal photographs and that multiple image classification models are applicable for use in veterinary medicine [69].

## 4. Discussion

Several challenges exist that are inherent to data sets associated with veterinary diagnostic imaging. Due to the nature of the patient caseload and the variability of the species and breeds encountered, the acquisition of large, uniform data sets can be challenging. Therefore, learning tasks must often be performed using small and often variable data sets. The lack of availability of examples of rare diseases for algorithm training is a limitation, meaning some diagnoses may be missed if such examples are not included in the training sets [2]. The availability of data sets will likely present one of the greatest challenges to the advancement of the use of AI in veterinary diagnostic imaging in the future, hence the need to develop large open-source data sets. Such data must also be curated in such a manner as to ensure ease of access and retrieval [1]. A number of additional challenges are also likely to present themselves in the future and will mirror those seen in the human medical fields, such as those associated with regulation and benchmarking of AI-related activities, as well as issues of privacy and a number of other ethical considerations such as culpability for misdiagnoses [1].

Despite an overall openness and enthusiasm to adapt and implement AI for use in human medical radiology, in general, a knowledge gap still exists that must be addressed before it can be fully adopted in veterinary medicine [74]. As the study by Boissady et al. (2020) showed, operators must be familiar and experienced with the use of AI as an aid, and trust in its results, in order to benefit from it [65]. Otherwise, failure to do so can add an error to the process. In addition, there is still a perception among a significant proportion of radiographers that AI could threaten or disrupt radiology practice, mainly due to a possible drop in demand or loss of respect for the profession [74,75]. It is likely that these perceptions also exist within veterinary diagnostic imaging and thus also present a hurdle to overcome before AI can be fully accepted within this profession in the future.

## 5. Conclusions

Although no reports exist in the veterinary literature, one logical next step for AI application in veterinary diagnostic imaging involves its use for the monitoring of lesion progression over time. Monitoring disease over time is essential not only for diagnosis and prognostic estimation but also for the evaluation of response to treatment. It consists of aligning diseased tissue across multiple diagnostic images taken over time, with the comparison of simple data to quantify change, for example, change in size, as well as variations in texture or heterogeneity computer-aided change analysis could detect subtle changes in characteristics not easily identified by the human eye and would also avoid the problems encountered with interobserver variability [1]. It is also likely that in the future, AI will play a greater administrative role, including patient identification and registration and medical reporting, and these advances are also likely to spill over into veterinary fields [1].

## Figures and Tables

**Figure 1 vetsci-09-00620-f001:**
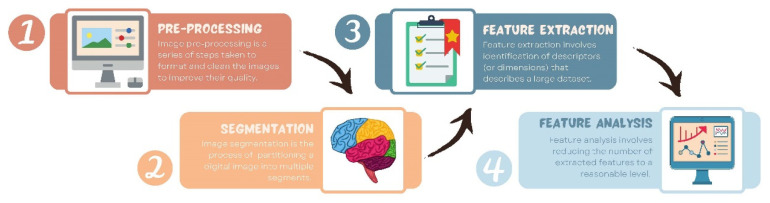
Illustration of the radiomics workflow steps: preprocessing, segmentation, feature extraction, and feature analysis.

**Figure 2 vetsci-09-00620-f002:**
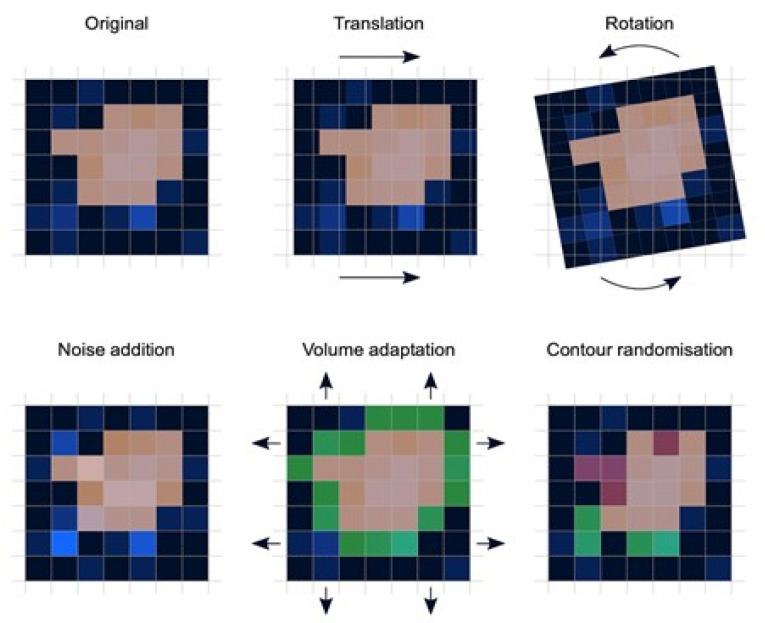
Variations in patient positioning, image acquisition, and segmentation affect each feature to varying degrees. If radiomic models use features that are not robust against such influences, they will perform poorly when applied to new data. Assessing feature robustness is thus recommended to improve the generalizability of radiomic models and feature analysis.

**Figure 3 vetsci-09-00620-f003:**
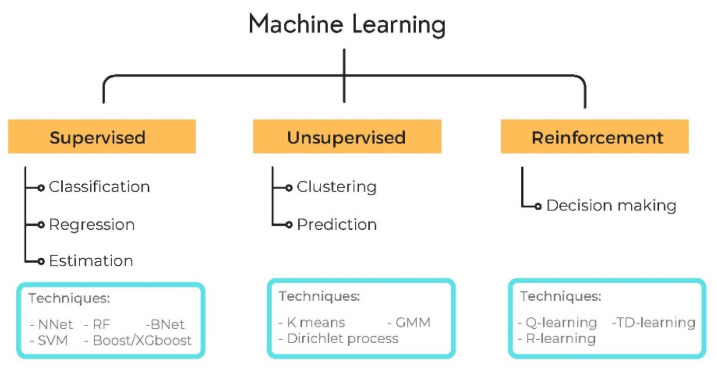
Supervised, unsupervised, and reinforced machine learning.

**Figure 4 vetsci-09-00620-f004:**
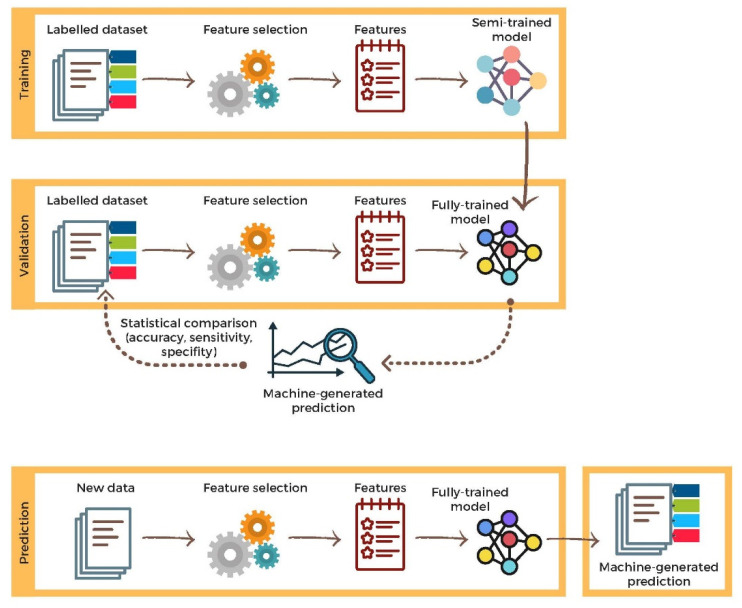
A brief description of supervised learning flowchart, including training, validation, and prediction.

**Table 1 vetsci-09-00620-t001:** Primary characteristics of publicly available open-source radiomics extraction tools.

	Programming Language	IBSI ^1^ Feature Definition	Full OS Compatibility	DICOM-RT ^2^ Import	Integrated Visualization	Radiomics Metadata Storage	Built-in Segme-Ntation	Reference
MITK	C++	No	Yes	Yes	No	No	No	Götz et al. 2019 [29]
MaZda	C++/Delphi	No	No	No	Yes	No	Yes	Szczypinski et al. 2009 [30]
PyRadiomics	Python	Yes	Yes	No	No	No	No	van Griethuysen et al. 2017 [31]
IBEX	Matlab/C++	No	No	Yes	Yes	Yes	Yes	Zhang et al. 2015 [32]
CERR	Matlab	Yes	Yes	Yes	Yes	Yes	Yes	Apte et al. 2018 [33]

^1^ ISBI “Image Biomarker Standard Initiative”. ^2^ DICOM-RT: “Digital Imaging and Communications in Medicine—Radiation Therapy”, an international standard to store, transmit, process, and display imaging data in medicine.

**Table 2 vetsci-09-00620-t002:** List of the most commonly used machine learning algorithms in medical imaging.

Model	Algorithm	Reference
*Linear Learning model*	Linear regression (LR) Principal component analysis (PCA) Linear discriminant analysis (LDA)	Nelder and Wedderburn (1972) [35] Jolliffe (2002) [36] Mclachlan (2004) [37]
*Nonlinear Learning model*	Logistic regression (LR) Naïve Bayes (NB) General additive models (GAM) Decision tree (DT) Support vector machine (SVM) Gradient boosting machine (Gboost) Advanced gradient boosting (XgBoost) Random forest (RF) Artificial neural network (ANN) K-nearest neighbors (K-NN) Deep learning (DL) PLS	Walker and Duncan (1967) [38] Russell (2003) [39] Hastie and Tibshirani (1990) [40] Quinlan (1987) [41] Cortes (1995) [42] Hastie, Tibshirani(1990) [40] Chen and Guestrin (2016) [43] Ho (1998) [44] Kleene (1956) [45] Fix and Hodges (1951) [46] Bishop (2006) [47], Schmidhuber (2015) [48]; LeCun, Bengio, and Hinton (2015) [49]; Goodfellow (2016) [50] Tibshirani (1996) [24]

**Table 3 vetsci-09-00620-t003:** Literature review of AI/radiomics studies in the veterinary imaging applications, with reported accuracies and conclusions. CNN: convolutional neural networks. N/A: not available.

Reference	Topic	Scale	Species	AI/Radiomic Algorithms	Accuracy	Conclusion
Basran et al., 2021 [63]	Lesion detection: equine proximal sesamoid bone micro-CT	Clinical N = 8 cases and 8 controls	Equine	Radiomics	N/A	Radiomics analysis of μCT images of equine proximal sesamoid bones was able to identify image feature differences in image features in cases and controls
Becker et al., 2018 [64]	Lesion detection: murine hepatic MRIs	Pre-clinical N = 8 cases and 2 controls.	Murine	Radiomics	N/A	Texture features may quantitatively detect intrahepatic tumor growth not yet visible to the human eye
Boissady et al., 2020 [65]	Lesion detection: canine and feline thoracic radiographic lesions	Clinical N = 6584 cases	Canine and feline	Machine learning - CNN	N/A	The described network can aid detection of lesions but not provide a diagnosis; potential to be used as tool to aid general practitioners
McEvoy and Amigo, 2013 [62]	Lesion detection: canine pelvic radiograph classification	Clinical N = 60 cases	Canine	Machine learning - CNN	N/A	Demonstrated feasibility to classify images, dependent on availability of training data
Yoon et al., 2018 [66]	Lesion detection: canine thoracic radiographic lesions	Clinical N = 3122 cases	Canine	Machine learning - CNN - BOF	CNN: 92.9–96.9% BOF: 79.6–96.9%	Both CNN and BOF capable of distinguishing abnormal thoracic radiographs, CNN showed higher accuracy and sensitivity than BOF
Banzato et al., 2018 [67]	Lesion characterization: MRI differentiation of canine meningiomas vs. gliomas	Clinical N = 80 cases	Canine	Machine learning - CNN	94% on post-contrast T1 images, 91% on pre-contrast T1-images, 90% on T2 images	CNN can reliably distinguish between different meningiomas and gliomas on MR images
D’Souza et al., 2019 [68]	Lesion characterization: assessment of B-mode US for murine hepatic fibrosis	Pre-clinical N = 22 cases and 4 controls.	Murine	Radiomics	N/A	Quantitative analysis of computer-extracted B-mode ultrasound features can be used to characterize hepatic fibrosis in mice
Kim et al., 2019 [69]	Lesion characterization: canine corneal ulcer image classification	Clinical N = 281 cases	Canine	Machine learning - CNN	Most models > 90% for superficial and deep corneal ulcers; ResNet and VGGNet > 90% for normal corneas, superficial and deep corneal ulcers	CNN multiple image classification models can be used to effectively determine corneal ulcer severity in dogs
Wanamaker et al., 2021 [70]	Lesion characterization: MRI differentiation of canine glial cell neoplasia vs. noninfectious inflammatory meningoencephalitis	Clinical N = 119 cases	Canine	Radiomics	Random forest classifier accuracy was 76% to differentiate glioma vs. noninfectious inflammatory meningoencephalitis	Texture analysis using random forest algorithm to classify inflammatory and neoplastic lesions approached previously reported radiologist accuracy, however performed poorly for differentiating tumor grades and types

## Data Availability

Not applicable.

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
