# Peer review of "A Review of Radiomics and Artificial Intelligence and Their Application in Veterinary Diagnostic Imaging"

_vetsci, 2022, doi:10.3390/vetsci9110620_

Round 1

Reviewer 1 Report

The present article discussed the employment of AI and radiomics in veterinary diagnostic imaging with highlighting the future directions in this track.  This is a hot topic in the field and an area where there is definitely a need to further extend research and applications. 

The article is well written, organized, and comprehensive. However, since the article concerns the applications in veterinary practice, more examples and details as scale of study (Preclinical/clinical), animal models, applications, etc...  are required in table 3. 

The English language and grammer need provisional improvement.

Author Response

Reviewer 1:

The present article discussed the employment of AI and radiomics in veterinary diagnostic imaging with highlighting the future directions in this track.  This is a hot topic in the field and an area where there is definitely a need to further extend research and applications. 

The article is well written, organized, and comprehensive. However, since the article concerns the applications in veterinary practice, more examples and details as scale of study (Preclinical/clinical), animal models, applications, etc...  are required in table 3. 

Table 3 has been amended accordingly.

Reviewer 2 Report

General comments:

I like the paper overall. There are two small typos, but mostly my reservations have to do with clarity of some of the technical explanations. I recommend some changes and expansions, all intended to improve clarity. I mention these in my detailed comments.

Detailed comments:

Page 2, lines 56-58: I like the clarity of this definition. I'm glad you included it.

Page 2, line 60: In American English, “which” is used instead of “that” in a subordinate clause like this. I think it may be the opposite in British English, but I make this comment in case the authors want to use American English.

Page 3, line 81: My only question here is whether you really need this level of detail to make your point.

Page 4, lines 122-123: By contrast to my previous concern, here I think it might be worth saying more.

Page 4, line 124: It is not clear how Figure 2 relates to the text. Is it part of pre-processing or segmentation?

Page 5, lines 144-147: Surely there is a clearer way of explaining “wavelet transformation.” Perhaps an example or a metaphor for the reader unfamiliar with this technique?

Page 5, line 156: I think most readers would want to know why “the application of artificial intelligence [is] ill-posed,” as they think computer-based AI could crunch through "big data" fairly efficiently. So I would urge you to explain this a bit.

Page 5, lines 158-159: "Redundant" I can understand, but how does either a human or AI actor decide what is relevant and what is not?

Page 5, line 167: Not sure what “Filter approach separates feature selection from classifier construction” means. I recommend that you try to explain this more plainly and clearly.

Table 1, line 1, second column: I suggest putting a footnote at the bottom of the table to remind the reader what the acronym IBSI means.

Table 1, line 1, column 4: I suggest putting a footnote at the bottom of the table here as well, to tell the reader what the acronym DICOM-RT means.

Page 9, line 210: It is not clear to what you are referring when you say “see below.” Is it Figure 4? Or something else?

Page 10, line 248: There’s a typo here. I think it should be “AI,” not “aI.”

Page 11, line 267: I do not understand why the text "Footnote 10" appears here.

Page 12, line 303-304: I find it surprising that the network had fewer errors than veterinarians supported by the network. It would be interesting to know why the veterinarians added error to the process.

Page 14, line 388: Table 3 is not referenced in the text. I think it would be good to find a place in the text to mention it.

Page 15, lines 395-396: Presumably this is why you earlier called for large open-source data sets....

Page 15, line 407: You may have covered this earlier, in your detailed descriptions of how radiographers have started to use AI, but it might be worth spelling out what this knowledge gap is.

Page 16, lines 420-421: You write, “It consists of aligning diseased tissue across multiple diagnostic images taken over time, with comparison of simple data to quantify change. For example, change in size, as well as variations in texture or heterogeneity.” The latter phrase is not a complete sentence, so I recommend that you change these two phrases as follows: “It consists of aligning diseased tissue across multiple diagnostic images taken over time, with comparison of simple data to quantify change, for example, change in size, as well as variations in texture or heterogeneity.”

Author Response

Reviewer 2:

General comments:

I like the paper overall. There are two small typos, but mostly my reservations have to do with clarity of some of the technical explanations. I recommend some changes and expansions, all intended to improve clarity. I mention these in my detailed comments.

Detailed comments:

Page 2, lines 56-58: I like the clarity of this definition. I'm glad you included it.

Page 2, line 60: In American English, “which” is used instead of “that” in a subordinate clause like this. I think it may be the opposite in British English, but I make this comment in case the authors want to use American English.

Following the reviewer’s suggestion, we have changed “that” to “which”.

Page 3, line 81: My only question here is whether you really need this level of detail to make your point.

Following the reviewer’s suggestion, we have reduced the level of detail, deleting lines 91-118 and replacing with a less detailed paragraph “Image acquisition, segmentation, intensity normalization, co-regulation, and noise filtering are three more analytical techniques that are crucial for processing the quantitative analysis of the images (Figure 2).”

Page 4, lines 122-123: By contrast to my previous concern, here I think it might be worth saying more.

A sentence has been added to line 131 “Deep learning (DL) is based on artificial neural networks where multiple layers (like neurons in human brain) are used to perform complex operations and extract higher levels of features from data. DL techniques require large amount of data and considerable computing resources to achieve the required accuracy.”

Page 4, line 124: It is not clear how Figure 2 relates to the text. Is it part of pre-processing or segmentation?

Following the reviewer’s suggestion, Figure 2 was moved to line 93 and linked with the additional sentence on lines 89-92.

Page 5, lines 144-147: Surely there is a clearer way of explaining “wavelet transformation.” Perhaps an example or a metaphor for the reader unfamiliar with this technique?

At the reviewer’s suggestion, the explanation of wavelet has been simplified. See line 149:

“For example, wavelet transformation is powerful tool for multiscale feature extraction. A wavelet is a function that resembles a wave. Scale and location are the basic characteris-tics of wavelet function. The scale defines how "stretched" or "squished" the function is whereas placement identifies the wavelet's position in time (or space). Decreasing the scale (resolution) of the wavelet can capture high frequency information and therefore, analyze well high-spatial frequency phenomena localized in space, and can effectively extract information derived from localized high-frequency signals.”

Page 5, line 156: I think most readers would want to know why “the application of artificial intelligence [is] ill-posed,” as they think computer-based AI could crunch through "big data" fairly efficiently. So I would urge you to explain this a bit.

Agreed: AI is very efficient, however if we have a large number of features and not enough data, we would not be able to get meaningful information (it is like if you a have a set of equations where the number of unknown variables is higher than the number of independent equations). So one way to avoid this issue, is to reduce the number of features to only those which are highly relevant.

We have added a sentence after line 156 to clarify “in particular if the number of data is not high”. 

Page 5, lines 158-159: "Redundant" I can understand, but how does either a human or AI actor decide what is relevant and what is not?

That is a very good question, and this is exactly why AI is powerful. The human does not decide, it is the algorithm that performs complex calculations to identify which features are highly connected to the outcome. In a recent article, one of the PIs of this article conducted AI analysis of diabetes data and reported that the present of some elements in blood can be a high indicator for prediabetes. This could not have happened without AI.

Page 5, line 167: Not sure what “Filter approach separates feature selection from classifier construction” means. I recommend that you try to explain this more plainly and clearly.

Following the reviewer’s suggestion, a sentence has been added to clarify this point (line 176):

“This implies that the machine learning algorithm handles the features removal and data classification in separate steps. As a result, the algorithm begins by picking out the most crucial features and eliminating the others, and then, in the second step, it only uses those features to classify the data. Wrapper approach measures the “usefulness” of features based on the classifier performance by using a greedy search approach that evaluates all the possible combinations of features against the classification-based evaluation criterion and keeps searching until a certain accuracy criterion is satisfied.”

Table 1, line 1, second column: I suggest putting a footnote at the bottom of the table to remind the reader what the acronym IBSI means.

Table 1, line 1, column 4: I suggest putting a footnote at the bottom of the table here as well, to tell the reader what the acronym DICOM-RT means.

Footnotes have been added to table 1 (line 194) as follows:

1ISBI “Image Biomarker Standard Initiative”

2DICOM-RT: “Digital Imaging and Communications in Medicine – Radiation Therapy”, an international standard to store, transmit, process and display imaging data in medicine.

Page 9, line 210: It is not clear to what you are referring when you say “see below.” Is it Figure 4? Or something else?

Line 226: have added “(see section 2.3)” where an explanation on normal ML versus DL (CNN and other methods) can be found.  

Page 10, line 248: There’s a typo here. I think it should be “AI,” not “aI.”

This has been corrected to “AI”, thank you.

Page 11, line 267: I do not understand why the text "Footnote 10" appears here.

This has been deleted, thank you.

Page 12, line 303-304: I find it surprising that the network had fewer errors than veterinarians supported by the network. It would be interesting to know why the veterinarians added error to the process.

True, this is an interesting point to note that we did not elaborate on. In fact, Boissady et al.(2020) noted that the access to the algorithm didn’t statistically improve the veterinarian’s error rate. They suggested that it might be due to the veterinarian’s lack of trust in the CNN’s pattern recognition and concluded that it could have been the result of lack of experience on behalf of the veterinarian using the AI as an aid.

A sentence to that effect has bee added (line 319): “It is interesting to note that the network failed to statistically improve the veterinarians’ error rate in this study, which the authors hypothesized could be due to lack of experience with use of AI as an aid, and failure to trust the CNN’s pattern recognition [47].”

Page 14, line 388: Table 3 is not referenced in the text. I think it would be good to find a place in the text to mention it.

References to table 3 have been added: lines 297, 308, 328, 342.

Page 15, lines 395-396: Presumably this is why you earlier called for large open-source data sets....

Indeed, see line 420, we have added “…hence the need to develop large open-source data sets.”

Page 15, line 407: You may have covered this earlier, in your detailed descriptions of how radiographers have started to use AI, but it might be worth spelling out what this knowledge gap is.

See line 429 additional sentence: “As the study by Boissady et al. (2020) showed, operators must be familiar and experienced with use of AI as an aid, and trust in its results, in order to benefit from it [47]. Otherwise, failure to do so can add error to the process.”

Page 16, lines 420-421: You write, “It consists of aligning diseased tissue across multiple diagnostic images taken over time, with comparison of simple data to quantify change. For example, change in size, as well as variations in texture or heterogeneity.” The latter phrase is not a complete sentence, so I recommend that you change these two phrases as follows: “It consists of aligning diseased tissue across multiple diagnostic images taken over time, with comparison of simple data to quantify change, for example, change in size, as well as variations in texture or heterogeneity.”

Upon the recommendation of the reviewers, the phrasing has been changed: “It consists of aligning diseased tissue across multiple diagnostic images taken over time, with comparison of simple data to quantify change, for example, change in size, as well as variations in texture or heterogeneity.”